# Exposure to violence affects the development of moral impressions and trust behavior in incarcerated males

Jenifer Z. Siegel [1,2], Suzanne Estrada[2], Molly J. Crockett [2,3] & Arielle Baskin-Sommers [2,3]

Individuals exposed to community violence are more likely to engage in antisocial behavior, resulting in a dramatic increase in contact with justice and social service systems. Theoretical accounts suggest that disruptions in learning underlie the link between exposure to violence and maladaptive behaviors. However, empirical evidence specifying these processes is sparse. Here, in a sample of incarcerated males, we investigated how exposure to violence affects the ability to learn about the harmfulness of others and use this information to adaptively modulate trust behavior. Exposure to violence does not impact the ability to accurately develop beliefs about agents' harm preferences and predict their choices. However, exposure to violence disrupts the ability to form moral impressions that dissociate between agents with distinguishable harm preferences, and subsequently, the ability to adjust trust behavior towards different agents. These findings reveal a process that may explain the association between exposure to violence and maladaptive behavior.

[1] Department of Experimental Psychology, University of Oxford, Oxford OX1 2JD, UK. [2] Department of Psychology, Yale University, New Haven, CT 06520, USA. [3]These authors contributed equally: Molly J. Crockett, Arielle Baskin-Sommers. Correspondence and requests for materials should be addressed to M.J.C. (email: molly.crockett@yale.edu) or to A.B.-S. (email: arielle.baskin-sommers@yale.edu)

Exposure to community violence, whether it is witnessing someone get chased or hurt, hearing gunshots in the neighborhood, or being directly chased, assaulted or shot at, is a significant public health concern. In the United States, over three-quarters of youth have been exposed to some form of community violence in their lifetime[1,2]. In general, both cross-sectional and longitudinal research finds that exposure to violence places young people at risk for persistent academic under-achievement[3], physical health problems (e.g., difficulty sleeping, headaches, heart disease, immune disease[4,5]), mental health problems (e.g., depression, anxiety, post-traumatic stress, anti-social personality[5–7]), and interpersonal problems (e.g., problems with trust, lower levels of empathy[8]). Additionally, individuals exposed to violence are more likely to engage in antisocial behavior[6,7], show earlier and more chronic aggressive behavior[9], and hold beliefs that can normalize or romanticize aggression[10]. As a result, exposure to violence dramatically increases the likelihood of involvement in the justice and social service systems[11].

Exposure to violence predisposes some individuals to diverse forms of negative life experiences and mental health problems, as well as comprises a prominent risk factor for a lifetime mired in aggression. Chronic exposure to violence, whether in a larger community or justice system context, shapes cognition in a way that is likely to distort perceptions of what is considered harmful behavior and how to react to harmful behavior. However, the precise social cognitive processes that may underlie these distortions in individuals exposed to violence is unclear. At the core of several theories about the relationship between exposure to violence and aggressive/antisocial behavior is the role of learning[10,12–15]. However, empirical evidence identifying and specifying the way in which learning is disrupted and can affect behavior in individuals exposed to violence remains elusive.

One aspect of learning that is especially relevant to adaptive social behavior is learning about whether other individuals might harm us. Harmfulness is a core dimension of moral character[16,17]. Research on social learning has shown that there are two, distinct, components of harmfulness learning. On the one hand, people use social cues to objectively update their beliefs about others' harmfulness by gradually accumulating information over time to predict future outcomes (i.e., in a Bayesian manner[18]). On the other hand, people form subjective impressions about moral character that emerge rapidly and effortlessly[19,20]. These beliefs and moral impressions are used to adaptively learn and decide whom to trust in social interactions[16,21]. For example, in a study by Siegel and colleagues[18], participants entrusted more money to agents who were less willing to harm others for profit and ascribed better moral character (subjective impression) to those agents compared to agents who were more willing to harm for profit. Together, these components of learning about other's harmfulness serve as powerful informational tools; for the purpose of survival, humans are evolutionarily inclined to identify potential foes and avoid them through adaptive social decision-making[22,23]. However, life experiences, such as exposure to violence that for some individuals follows them continuously from the community to prison[24], are likely to shape social learning and resulting social behaviors. Prior research linking exposure to violence to normalized views of aggression and aberrations in interpersonal functioning raises the possibility that exposure to violence may impact learning about the harmfulness of others, and by extension, behaviors that rely on trust. To date, however, there has been no research on exposure to violence and harm learning.

To examine the relationships among exposure to violence, harm learning, and trust behavior, we administer a harmfulness learning task[18] to a sample of incarcerated males. While a sample of currently incarcerated individuals is not the same as a sample

from the general population, this type of sample does serve as an informative sample in which to explore how differences in exposure to violence impact harm learning. It is well-documented that exposure to violence among the incarcerated covers the full continuum of potential experiences compared to the general population where scores are often restricted in range and narrowly centered around a few points within that range[25–27]. Moreover, by studying a sample of currently incarcerated individuals, we are better poised to investigate the variation in exposure to violence within a sample that is already demonstrating the theorized behavioral effects of such exposure.

In the task, participants predict and observe the choices of two agents who repeatedly decide whether to inflict painful electric shocks on another individual in exchange for money (Fig. 1a). The two agents substantially differ in their preferences towards harm (i.e., their exchange rate between money and pain). On each trial, participants predict the choice made by the agent and receive immediate feedback about their accuracy. Every three trials, participants rate their overall subjective impression of the agent's moral character (on a scale from nasty to nice) and their certainty of that impression. This task enables us to measure two distinct components of harm learning: the ability to develop accurate beliefs about the agents' objective exchange rates between money and pain (a quantity that is used to predict their choices), and the use of those estimates to form subjective, global impressions about other's moral character. After the harmfulness learning task, participants engage in a one-shot trust game[28] with each of the agents. All participants complete a battery that assesses exposure to violence using the Exposure to Violence Scale (ETV)[29], as well as a clinical assessment measuring different aspects of antisociality to address potential confounds. We show that the ability to learn about harmfulness is not affected by exposure to violence. However, exposure to violence impairs the development of subjective impressions, and consequently, the ability to adapt trust behavior toward more harmful vs. less harmful agents.

## Results

**Beliefs and predictions about harm preferences**. We first investigated participants' ability to develop accurate beliefs about the agents' objective harm preferences and predict their decisions. On average, participants predicted accurately 72% of the good agent's choices and 77% of the bad agent's choices. There was no relationship between ETV score and prediction accuracy for either agent (Spearman's $\rho$, good: $\rho = -0.065$, $p = 0.483$; bad: $\rho = 0.043$, $p = 0.639$). This suggests that participants with higher exposure to violence were equally motivated to learn the harm preferences of the agents, relative to those with lower exposure to violence.

Next, we examined how rapidly participants updated their beliefs about the agents' preferences in response to feedback. To this end, we fit a hierarchical Bayesian model for learning stable preferences under conditions of uncertainty to participants' predictions. The model defines how beliefs about an agent's harm preference evolve over time as a function of a participant-specific parameter $\omega$, capturing inter-individual differences in the rate of belief updating (see Methods and Fig. 1b)[30]. Formal model comparison indicated that the hierarchical Bayesian model outperformed two alternative Rescorla Wagner models in our sample of participants (see Methods and Supplementary Note 1). Replicating previous research[18], beliefs about the bad agent's preferences were more rapidly updated in response to feedback, as indicated by a higher $\omega$, than beliefs about the good agent's preferences (signed rank test, $Z = -2.328$, $p = 0.020$; Fig. 2a). ETV score was not significantly related to $\omega$ for either agent

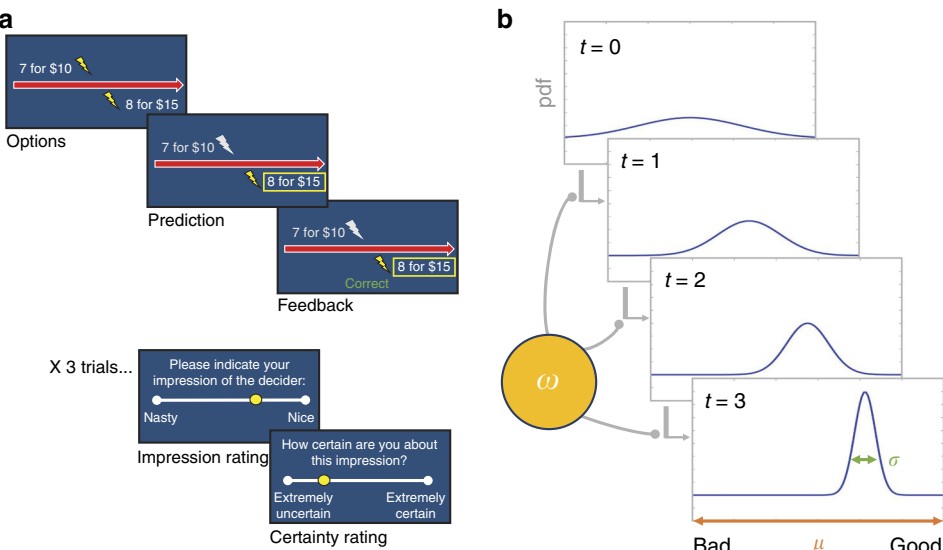

**Fig. 1** Learning task and model. **a** Representation of the task schematic, created by the authors. Participants predicted sequences of choices for two agents (Decider A and Decider B). On each trial the agent chose between two options: more shocks inflicted on another person in exchange for more money, or fewer shocks for less money. After making their prediction, participants observed the agent's actual choice along with feedback indicating whether they were correct or not in their prediction. Every third trial participants made a judgment about the agent's moral character (ranging from nasty to nice) and indicated how certain they were about their judgment. **b** Model schematic for learning about a good agent, modified from Siegel et al.[18]. Beliefs about the agent's harm preference are represented by probability distributions with a mean $\mu$ and variance $\sigma$. Beliefs evolve over time as a function of Gaussian random walks whose step-size is governed by $\omega$, a participant-specific parameter that captures individual differences in the rate at which beliefs evolve over time, $t$

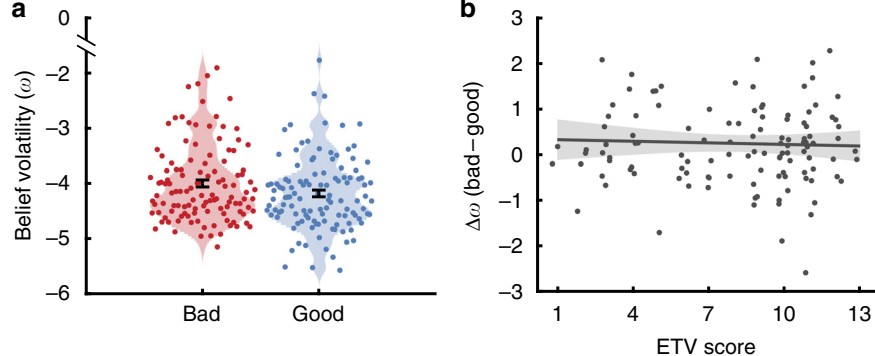

**Fig. 2** Objective harm learning does not covary with exposure to violence. **a** Beliefs about the bad agent's harm preferences were more volatile than beliefs about the good agent's harm preferences. **b** Between-agent asymmetries in belief updating ($\Delta\omega$ = bad agent belief volatility − good agent belief volatility) were not related to participant's ETV score, suggesting that exposure to violence does not significantly impact the underlying processes of objective harm learning. Error bars represent standard error of the mean. Error bands represent 95% confidence intervals. Source data are provided as a Source Data file

(Spearman's $\rho$, good: $\rho = 0.025$, $p = 0.785$; bad: $\rho = -0.014$, $p = 0.879$), nor was it related to the difference in $\omega$ between good and bad agents ($\Delta\omega$: $\rho = 0.008$, $p = 0.929$; Fig. 2b). Together, these results suggest that objective harm learning was largely intact in this sample and did not covary with exposure to violence.

**Subjective impressions of the agent's moral character**. Despite the fact that ETV score did not impact learning the objective features of agents' preferences, we observed a strong effect of ETV score on participants' subjective global impressions of the agent's moral character. A robust linear regression was used to predict subjective moral impression ratings as a function of agent (good vs. bad), ETV score, and their interaction. We report means and standard error of the mean (sem) as mean ± sem. We included an additional regressor to control for trial number, and found no effects on impression ratings ($\beta = 0.003 \pm 0.003$, $t = 0.955$, $p = $

0.340). Replicating previous research, in general, participants formed more favorable impressions of the good agent's moral character than the bad agent's moral character ($\beta = -1.300 \pm 0.075$, $t = -17.284$, $p < 0.001$). Higher ETV scores predicted more negative impressions of the good and bad agents' moral character ($\beta = -0.018 \pm 0.006$, $t = -2.902$, $p = 0.004$). There was a significant interaction between ETV score and type of agent ($\beta = 0.037 \pm 0.009$, $t = 4.222$, $p < 0.001$), indicating that for participants with higher ETV scores, there was less differentiation in their impressions of the good and bad agents' moral character (Fig. 3a).

To further investigate the interaction between type of agent and exposure to violence on subjective impression ratings, we ran separate regressions on ratings for the good and bad agent. Specifically, we asked whether diminishing effects of agent with higher ETV scores were driven by the good agent, the bad agent, or both. These analyses revealed that the effects were not

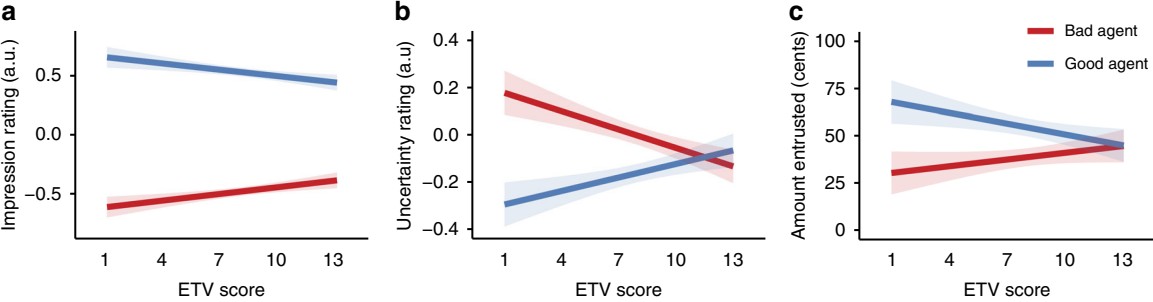

**Fig. 3** Model estimates showing diminishing effects of agent with increasing exposure to violence. Participants with higher ETV scores showed less differentiation in their subjective impressions of good (blue line) vs. bad (red line) agent's moral character (**a**) and reported smaller discrepancies in the uncertainty of their impressions of good and bad agents (**b**). Higher ETV scores also resulted in smaller discrepancies in the amounts that participants entrusted with good vs. bad agents in a one-shot trust game (**c**). Y-axis in figures **a** and **b** denote standardized values (z-scored). Error bands represent 95% confidence intervals. Source data are provided as a Source Data file

specifically driven by either agent alone. Higher ETV scores predicted more favorable impressions of the bad agent ($\beta = 0.019 \pm 0.007$, $p = 0.004$) and less favorable impressions of the good agent ($\beta = -0.019 \pm 0.006$, $p = 0.001$; see Fig. 3a).

Of note, participants with higher ETV scores were no more likely to predict worse harm intentions of the agents in the task before they observed any of their choices (Spearman's $\rho$, $\rho = -0.082$, $p = 0.361$) and were no less trusting of others in general (as indicated by scores on the General Trust scale[31]; $\rho = -0.040$, $p = 0.665$). These findings suggest that exposure to violence affects how participants in this sample form subjective impressions about other's moral character through observing their choices, rather than affecting prior beliefs about others.

**Certainty of subjective impressions.** Past work in non-incarcerated samples indicates that adults hold more certain positive impressions of others and more uncertain negative impressions, which is hypothesized to serve the adaptive social function of enabling people to more easily update negative impressions that turn out to be inaccurate[18,32]. Consistent with previous research, uncertainty decreased over time as participants were exposed to more information about the agents' harm preferences (Robust regression, $\beta = -0.018 \pm 0.003$, $t = -5.969$, $p < 0.001$). Furthermore, participants expressed greater uncertainty in their impressions of the bad agent, relative to the good agent ($\beta = 0.513 \pm 0.078$, $t = 6.605$, $p < 0.001$).

To investigate whether exposure to violence affected participants' uncertainty in their impressions of the agents' moral character, we performed a robust linear regression to investigate the effects of agent, ETV score, and their interaction on participants' ratings of uncertainty about their impressions of the agents. Participants with higher ETV scores were more uncertain in their impressions overall ($\beta = 0.019 \pm 0.006$, $t = 2.982$, $p = 0.003$), and this interacted with the effect of agent ($\beta = -0.045 \pm 0.009$, $t = -4.973$, $p < 0.001$; Fig. 3b). Consistent with our findings on subjective impressions, the interaction indicated that participants with higher ETV scores expressed smaller differences in their uncertainty ratings between good and bad agents, such that they became more uncertain that the good agent was good, and less uncertain that the bad agent was bad. Notably, smaller differences in impression ratings between good and bad agents predicted less discrepant uncertainty ratings (Spearman's $\rho$, $\rho = 0.336$, $p < 0.001$). Finally, mirroring the subjective impression results, these effects were not specifically driven by either agent alone. Higher ETV scores predicted less uncertain impressions of the bad agent ($\beta = -0.025 \pm 0.007$, $t = -3.795$, $p < 0.001$) and more uncertain impressions of the good agent ($\beta = 0.019 \pm 0.006$, $t = 3.126$, $p = 0.002$; see Fig. 3b). Results from our robust linear regressions did not change after controlling for age and education (see Supplementary Tables 4, 5).

**Trust behavior.** Although exposure to violence impaired participants' ability to form distinct subjective impressions of agents with different harm preferences, it is unclear whether this has consequences for social behavior. To address this question, we asked participants to engage in a one-shot trust game with each of the agents, after predicting all the agents' choices in the harm learning task (see Methods). Previous work has shown that non-incarcerated adults adjust their behavior in the trust game according to the harm preferences of the agent with whom they are interacting (i.e., people entrust significantly less money with those who treat others poorly than those who treat others well[18,33]).

To investigate whether adaptive trust behavior was diminished in participants with higher ETV scores, we entered the amount participants entrusted in a repeated measures general linear model with agent (good vs. bad) as the within-subject factor and ETV score as a continuous covariate. Consistent with previous research, participants entrusted more points with the good agent than the bad agent ($F(1,119) = 6.202$, $p = 0.014$, $\eta^2 = 0.056$). The effect of agent was significantly moderated by ETV score ($F(13,119) = 2.142$, $p = 0.017$, $\eta^2 = 0.210$; Fig. 3c). The interaction indicated that higher ETV scores predicted smaller discrepancies in the amount participants entrusted with the good vs. the bad agent. Specifically, those with higher ETV scores entrusted significantly less with the good agent (Spearman's $\rho$, $\rho = -0.220$, $p = 0.016$), and consequently ended up earning fewer points overall ($\rho = -0.325$, $p < 0.001$). ETV scores did not significantly affect the amount that participants entrusted with the bad agent ($\rho = 0.119$, $p = 0.198$). Thus, exposure to violence was associated with maladaptive trusting behavior, specifically when interacting with those who are less willing to harm others, and this had a negative impact on their overall earnings.

Given the relationship between exposure to violence and differential trust behavior (Δentrust, calculated as amount entrusted with good agent − amount entrusted with bad agent), it is possible that participants' final subjective impressions of the agents' moral character (Δjudgment, calculated as final impression of the good agent – final impression of the bad agent) account for (i.e., mediate) that relationship. We found a significant indirect effect of Δjudgment on Δentrust, effect = −0.812, CI [−1.705, −0.054] (see Supplementary Note 2 for full mediation results and additional analysis), suggesting that impressions about other's moral character account for differences in social behavior among participants with higher levels of exposure to violence.

Despite these results, some might question the validity of the trust game in the current sample given their incarceration status. Therefore, we examined whether the extent to which participants adjusted their trust behavior according to the agents' harm preferences predicted social behavior in prison. Less discrepant behavior towards good and bad agents was associated with more behavioral violations in prison (Spearman's ρ, $\rho = -0.208$, $p = 0.023$), and specifically with aggressive violations against persons ($\rho = -0.217$, $p = 0.020$). This suggests the ability to adjust trust behavior based on impressions of other's moral character, as measured by our task, captures variance in real-world social behavior.

However, it's possible that this relationship between less discrepant behavior towards good and bad agents and more behavioral violations in prison is largely explained by the relationship with ETV scores. Indeed, higher ETV scores predict more behavioral violations in prison (Spearman's ρ, $\rho = 0.450$, $p < 0.001$). However, we predicted that this relationship would be mediated by the extent to which participants differentiated in their subjective impressions and trust behavior between the good and bad agent. Consequently, we applied a serial multiple mediation analysis using the PROCESS macros for SPSS[34] (model 6) that allowed us to determine the causal link between mediators with a specified direction of causal flow. We investigated whether the relationship between exposure to violence and prison violations was mediated by trust behavior (Δentrust) as a function of impression sensitivity (Δjudgment). ETV score was only a marginally significant predictor of prison violations after impression sensitivity and trust behavior were accounted for (direct effect = $0.622 \pm 0.340$, $p = 0.070$). The indirect effects were tested using a bootstrap estimation approach with 5000 samples. These results indicated the indirect serial coefficient was significant (indirect effect = $0.099 \pm 0.071$, 95% CI = [0.002, 0.274]; see Supplementary Note 3 for full mediation results and Supplementary Fig. 1), suggesting that disruptions in the ability to form distinguishable impressions resulting from higher ETV scores, translates into maladaptive trust behavior, which in turn leads to a greater number of direct violations in prison.

**Specificity of exposure to violence effects**. Previous work has shown that exposure to violence is associated with antisocial behavior and psychopathic traits[10,35–37]. Here we also found that ETV score was associated with increased antisociality, as indicated by higher scores on the Hare Psychopathy Checklist-Revised[38] (PCL-R) and increased symptoms of Antisocial Personality Disorder[39] (APD) (Spearman's ρ, PCL-R total score: $\rho = 0.394$, $p < 0.001$; APD symptoms: $\rho = 0.261$, $p = 0.004$). This leaves open the question of whether the effects of exposure to violence on subjective impressions observed here are a primary consequence of exposure to violence, or an indirect consequence of possessing characteristics that predispose exposure to violence, such as Psychopathy or Antisocial Personality Disorder. To assess whether ETV score had a direct, as opposed to indirect, effect on subjective impressions and uncertainty ratings, and social behavior, we entered each covariate (PCL-R total score and total number of APD symptoms) separately into our regressions with ETV score. Across all measures, we found that interactions between agent and ETV score remain significant even when we include the interaction between agent and each covariate (see Supplementary Tables 4–6 for all analyses including covariates). An alternative possibility is that the observed effects of exposure to violence on impressions reflect a general impact of traumatic experiences, rather than being specific to community violence. To investigate, we entered scores from the Childhood Trauma Questionnaire[40] (CTQ) into our regression with ETV score.

Again, we found that the interactions between agent and ETV score remain significant even after controlling for CTQ (see Supplementary Tables 4–6 for full analysis). Together, this suggests that being exposed to violence had a direct effect on subjective impressions of moral character and social behavior and that findings could not be entirely explained by antisocial psychopathology or childhood trauma.

## Discussion

The ability to infer other's intentions and predict their behavior is crucial for successful social interactions. In particular, learning whether others are likely to harm us is important for consequential social decisions like deciding whom to trust. However, there are environmental experiences that may impact how we learn about harm and use this information to make adaptive social decisions. Exposure to violence is one environmental experience that is associated with aberrations in beliefs about harm[10,15]. As a result, exposure to violence is related to behaviors that reflect a lack of trust and prosociality (e.g., aggression, crime), increasing contact with systems of social control.

The current data suggest that, in a sample of currently incarcerated males, exposure to violence adversely impacts some components of harm learning, but not all. Participants with higher ETV scores showed an ability to develop accurate beliefs about others by objectively encoding their harm preferences. However, exposure to violence appeared to disrupt the formation of subjective, global impressions of other's moral character from observed harm behavior. Participants with higher ETV scores formed more positive and less uncertain impressions of harmful agents and more negative and less certain impressions of helpful agents. Moreover, these differences in subjective impressions associated with higher ETV scores led to maladaptive trust behavior, such that participants with higher ETV scores extended less trust than optimal when interacting with a good agent. Finally, the link between exposure to violence and maladaptive trusting behavior was mediated by the disturbances in impression formation. In turn, this led to significantly more violations in prison, suggesting that the effects of exposure to violence on real social behavior in prison is predicted by subjective impressions and trust behavior as measured by our task. On the whole, these findings raise the intriguing possibility that exposure to violence does not fundamentally disrupt all components of social learning, but instead may produce a problem with generating global subjective social impressions and translating those impressions into adaptive social decision-making.

Our findings are consistent with evidence that the ability to learn the value of information is cognitively and neurally distinct from the ability to use learned information to guide decision-making and behavior[41,42]. Cognitively, participants with higher ETV scores were able to learn harm preferences of different agents but formed subjective impressions that appeared to normalize beliefs about harm in the bad agent[10,15], seeing that agent as more similar to the good agent. This dissociation between learning and using learned information is consistent with neural lesion studies show that the lateral orbitofrontal cortex (OFC) is associated with learning the value of stimuli, whereas the medial OFC is associated with translating stimulus value representations into decisions[43]. Although there are no imaging studies that directly examine exposure to violence, studies looking at the combination of physical abuse and adversity (e.g., seeing intimate-partner violence, bullying, physical and sexual abuse) note structural and functional abnormalities in the OFC[44–46], and individuals (both incarcerated and non-incarcerated) prone to aggressive and antisocial behavior also display abnormal medial OFC structure and function[47–50]. Taken together, findings from

these parallel literatures suggest that diminished use of learned information may be a consequence of OFC dysfunction in individuals exposed to violence.

An especially unfortunate consequence of disrupted use of harm learning may be a pervasive inability to develop healthy social relationships with trustworthy individuals and a greater likelihood of placing trust in the wrong people. Consistent with findings in the present study showing the impact of subjective impressions on trust behavior, research from the fields of sociology, psychology, and economics highlight that individuals who reside in communities with high rates of crime and disorder experience mistrust in their interactions with strangers, prosocial members of their community, and institutions[51–53]. Justice-involved individuals tend to reside in these types of communities characterized by crime and disorder, where social interactions and systems that may provide pathways out of serious and persistent offending are absent, severely debilitated, or sparse[51,54]. Moreover, once caught within the justice system, it is likely that these types of interactions are reinforced through new exposures to violence and negative social interactions[24]. Combined, community context and justice-involvement translates to lower access to informal and formal resources, homebased learning, and chronic re-exposure to violence[55,56]. The resulting pattern is that some individuals, like justice-involved individuals, are more likely to live in communities of unrelenting social and economic deprivation. These environmental characteristics do not solely impact the incarcerated but spills over to other community members: those who are incarcerated are released and their behavior and experiences impact family members, social acquaintances, and strangers in their own communities. For that matter, the disproportionate presence of incarcerated individuals in disadvantaged communities is not seen as aberrant but is often just part of living in these communities[57]. Thus, the combination of environmental characteristics and disruptions in the cognitive processes at the individual level are critical for the development and maintenance of trust and may ensnare individuals in a trajectory that continually reinforces maladaptive social connections, ultimately limiting chances for economic stability[58–60] and psychosocial wellbeing[5].

Before concluding, methodological and conceptual limitations should be noted. The present sample is limited to incarcerated offenders, and thus we do not know whether or how incarceration status may impact the relationship between exposure to violence and harm learning. However, it is important to note that all task main effects replicated previous findings in non-incarcerated samples. For instance, previous work using the same task has shown that people form less positive, more uncertain, and more volatile beliefs about the bad agent, relative to the good agent, and adjust their trust behavior according to the harm preferences of the agents[18]. We observe the same pattern of results in our sample of incarcerated individuals. Moreover, length of incarceration (see Supplementary Tables 4–6) and other correlates known to increase risk for incarceration did not impact the reported exposure to violence effects. Ultimately, being currently incarcerated is just one type of adverse outcome related to exposure to violence that should not be seen as excluding the importance of the lived experience of exposure to violence for these individuals[61–65].

While it may be useful to replicate the findings in a sample of non-incarcerated individuals, this raises important experimental considerations. From a scientific perspective, using a sample with sufficient variability in ETV scores, and whose experience with exposure to violence has led to great personal cost, is essential. Notably, the distribution of ETV scores in our sample of incarcerated individuals covers the full range of the scale. Endeavors in samples typical of psychology research, such as university or crowdsourced samples, often suffer from restricted range in ETV scores. Nonetheless, to test for generalizability, future research should replicate the present findings in a sample of non-incarcerated individuals whose ETV scores are reflective of a range of experiences.

A final consideration is that implementing shocks in the harmfulness learning task is not as extreme a behavior as what might be seen in the real world (e.g., sexual assault, murder) for individuals exposed to violence or involved in the justice system. Therefore, it is possible that the objective learning of other's harm preferences could be different with more extreme behaviors. Future research should continue to investigate components of learning in those exposed to violence and vary the stimuli used to assess learning that consider cultural and situational contexts.

The relationship between exposure to violence and negative life experiences is undeniable. However, an understanding of how this environment shapes cognition and behavior is less clear. The present study identifies a specific deficit in the ability of incarcerated individuals exposed to violence to adapt social behavior towards agents with distinguishable harm preferences. Continuing to identify and specify the processes that are altered by exposure to violence will be crucial for understanding how individuals experience, incorporate, and react to their particular social environment.

## Methods

**Participants.** The present sample included 119 males from a high-security correctional institution in Connecticut. We used a prescreen of institutional files and assessment materials to exclude justice-involved individuals who: were not between the ages of 18 and 75; scored below 70 on a brief measure of IQ (Shipley Institute of Living Scale[66]) performed below the fourth-grade level on a standardized measure of reading (Wide Rage Achievement Test-III[67]) had diagnoses of schizophrenia, bipolar disorder, or psychosis, not otherwise specified; were currently taking psychotropic medication; or had a history of medical problems (e.g., uncorrectable auditory or visual deficits, head injury with loss of consciousness greater than 30 min, seizures, neurological disorders) that may impact their comprehension of the materials. The Yale University Human Investigation Committee approved the procedures used in the present study. The study complied with all relevant ethical regulations for work with human participants and all participants provided written informed consent. See Supplementary Table 1 for participant demographic information.

**Harmfulness learning task.** In the task, participants predicted a sequence of 50 choices made by each of two agents (100 choices total). For each choice, the agent chose between two options: more money for themselves plus more shocks for an anonymous other individual, or less money for themselves plus fewer shocks for the other individual (Fig. 1a). For each trial, participants received feedback about their accuracy. No a priori information was provided about the agents; thus, optimal behavior required participants to learn the agents' preferences over time. Participants predicted all choices for one agent at a time, and the order of agents was randomized across participants.

Following every third prediction, participants indicated their current impression of the agent's moral character on a continuous visual analogue scale rating from 0 (nasty) to 100 (nice)[68] and indicated how certain they were about their impression on a scale ranging from 0 (extremely uncertain) to 100 (extremely certain) (see Fig. 1a). Together, this provided us with a trajectory of participants' subjective impression ratings of each agent's moral character and how certain participants were about their characterization.

To manipulate harm preferences, we created one agent who was more averse to harming others (good agent) and one agent who was less averse (bad agent). This was operationalized as their exchange rate between money for themselves and shocks for the other individual, with the good agent requiring more money per shock inflicted than the bad agent (good agent = $2.40 per shock; bad agent = $0.43 per shock). The agents selected from identical choice sets; however, because the agents had different preferences towards harm, they behaved differently 50% of the time. For details on how the trial sequences were created and the agents' behavior simulated, see Supplementary Methods.

To incentivize participants to learn about the preferences of the agents, before beginning the task they were instructed to pay close attention to the deciders and learn about their behavior, because they will interact with each of them in a computerized trust game at the end of the study which could earn them reward points.

**Trust game**. After completing the harmfulness learning task, participants played a trust game with each of the agents. In the game, participants were endowed with 100 points that they could entrust with each agent. Any amount that they entrusted with the agent would be tripled, and the agent could then choose how much of the tripled amount to return to the participant. We instructed participants that the percent returned by each agent had been predetermined, and thus the agents were not playing actively. We set the returned amount to correspond to the agents' actual harm tendencies, such that the good agent behaved less selfishly than the bad agent and therefore returned a larger proportion of the entrusted points. The final number of points was tallied and the top five earners were added to a leader board that was on display to all study participants in the testing room (Note: The Connecticut Department of Correction did not allow researchers to pay justice-involved individuals). Of particular interest was the difference in amount entrusted with the good agent vs. the bad agent (Δentrust = amount entrusted with the good agent – amount entrusted with the bad agent).

**Exposure to Violence Scale**. The ETV scale[29] was used to measure lifetime exposure to violent events. The questionnaire consisted of 13 items, documenting the types of both experienced and observed violence (e.g., "Have you been hit, slapped, punched, or beaten up?" and "Have you seen someone else get attacked with a weapon, like a knife or bat?"). Participants were asked to respond to each item based on a dichotomous choice (*yes/no*). If *yes* was selected, participants indicated the number of times they experienced this situation in their lifetime. The two scales, experienced and observed, showed moderate overlap (Spearman's ρ, $\rho = 0.607$, $p < 0.001$). Thus, we examined a total exposure to violence score using a sum of all 13 items. Internal consistency for ETV total score was 0.86. ETV scores were normally distributed (skewness: −0.605, kurtosis: −0.810). Ninety-nine percent of the sample reported experiencing at least one exposure to violence in their lifetime and ~30% of the sample reported experiencing over nine (the median) different exposures to violence in their lifetime. Lifetime frequency of exposure to violence ranged from two times to 11,465 times (median = 88).

**General Trust Scale**. The General Trust scale[31] was used to measure general beliefs about the honesty and trustworthiness of others. Participants were asked to indicate to what extent they agree (1) or disagree (5) with six statements (e.g., "Most people are trustworthy"). The scores from each statement were averaged together to produce a continuous measure of generalized trust.

**Hare psychopathy checklist revised (PCL-R)**. The PCL-R[66] used information gleaned from a life-history interview and a review of institutional files to score participants on the presence of 20 different items (e.g., superficial charm, shallow affect, impulsivity, criminal versatility). A score of 0, 1, or 2 was given for each item according to the degree to which a characteristic was present. PCL-R total scores ranged from 0 to 40. The reliability and validity of the PCL-R has been well established[38,69]. Inter-rater reliability for 24% of the sample was 0.991 (alpha).

**Antisocial personality disorder**. Participants were assessed for Antisocial Personality Disorder (APD) during a semi-structured diagnostic interview. The interview evaluated the age and frequency of engagement in behaviors outlined in the Diagnostic Statistical Manual-5[39] (DSM). A diagnosis of APD was given if there was evidence of conduct disorder (CD) prior to age 16 and sufficient adult anti-social symptoms (e.g., aggression, irresponsibility). Inter-rater reliability for 32% of the sample was 0.989 (Cohen's kappa).

**Childhood Trauma Questionnaire**. We used the Childhood Trauma Questionnaire (CTQ[40]), a 28-item questionnaire, to assess maltreatment experiences prior to age 18. It consisted of five clinical scales: emotional abuse, physical abuse, emotional neglect, and sexual abuse. Items were rated on a 5-point Likert-type scale with response options ranging from *Never True* to *Very Often True*. For the present study, the total score was examined. For this sample, the total score demonstrated good internal consistency (Cronbach's $\alpha = 0.824$).

**Computational model**. We compared three different computational models to describe how participants learned the agents' preferences and predicted their choices. First, we fit a Hierarchical Gaussian Filter model[30,70] (HGF), which identified participant-specific parameters to describe each participant's learning process. Beliefs about an agent's harm preference were updated using a Bayesian reinforcement learning algorithm, with precision-weighted prediction errors driving belief updating at the different levels of the hierarchical model. Second, we fit a Rescorla Wagner model, in which beliefs were updated by prediction errors with a fixed learning rate. Third, we fit a modified Rescorla Wagner model, in which beliefs were updated by prediction errors with separate fixed learning rates for helpful and harmful outcomes. All model fitting was implemented using the HGF toolbox (https://tnu.ethz.ch/tapas). For a full list of priors, see Supplementary Table 2. For further details about each model see Supplementary Table 3.

As in previous studies[18], formal model comparison indicated that the HGF model outperformed the two alternative Rescorla Wagner models in our sample of participants (see Supplementary Note 1 for details). The HGF model generated a trial-wise sequence of belief estimates about each agent's harmfulness (i.e., the exchange rate between money and pain, latent variable, $\mu$; a trial-wise sequence of uncertainties on those beliefs (latent variable, $\sigma$); and a global estimate of belief volatility (parameter, $\omega$) that describes the rate at which beliefs evolve over time (Fig. 1b). Belief volatility is set in log space and is monotonically related to belief uncertainty (i.e., more uncertain beliefs are more volatile[30]; for example, a change in ω from −3.5 to −4.0 corresponds to a 20% decrease in the average variance of posterior beliefs, σ.) In short, the model describes trial-wise updating of beliefs about the agent's preferences towards harm, which approximates Bayes optimality (in an individualized sense given differences in ω) and determines the participant's estimate of the probability that an agent will harm.

**Statistical analyses**. All data analysis was completed in Matlab (Mathworks) and PASW Statistics 24 (SPSS/IBM). All statistical tests were two-sided. We used robust linear regression models with a bisquare weighting function to analyze the z-scored trial-by-trial rating data (impression and certainty ratings). We used nonparametric statistical tests that do not make any assumptions about the underlying distributions of variables (e.g., Spearman's ρ and signed rank tests). To investigate whether the relationship between ETV score and differences in social behavior were mediated by differences in participants final harm judgments of the agents we used the PROCESS macro for SPSS[34].

## Data availability
The data that support the findings of this study are available from the corresponding author upon reasonable request. Source data for Figs. 2 and 3 are provided with the paper.

## Code availability
All relevant Matlab code are available from the corresponding author upon request.

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

## Acknowledgements

We thank those affiliated with the Connecticut Department of Correction, particularly Warden Scott Erfe and Dr. Patrick Hynes for their continued support of this research; and the research assistants who helped collect these data. This work was supported by a Clarendon and Wellcome Trust Society and Ethics award (104980/Z/14/Z), a Wellcome Trust ISSF award (204826/Z/16/Z), and the Academy of Medical Sciences (SBF001/1008).

## Author contributions

J.Z.S., M.J.C. and A.B-S designed the experiment. A.B-S. and S.E. collected the data. J.Z.S. analyzed the data. J.Z.S., A.B.S. and M.J.C. wrote the article with S.E. providing critical revisions.

## Additional information

**Competing interests:** The authors declare no competing interests.

