## [Peer Review File · Nature Communications]

Reviewers' comments:

Reviewer #1 (Remarks to the Author):

This manuscript reports a study of moral judgment and behavior in incarcerated males with varying degrees of exposure to violence. The basic finding is that, among this population, exposure to violence is unrelated to the ability to learn facts about others' willingness to harm, and yet with increasing exposure to violence there is a decreasing ability to use these facts to update trait impressions or guide trust behavior. The research questions are interesting, the experimental approach is sound, and the exposition is clear. Although my general feelings are positive, I do have a few substantial concerns.

1. The main theoretical conclusion is that "exposure to violence does not fundamentally disrupt learning processes, but instead may produce a problem with generating global social impressions based on learned information and translating these impressions into adaptive decision-making." (367-370) The most direct statistical test of this hypothesis would be to show that the correlation between "learned harmfulness" and "trait information and/or trust behavior" is moderated by ETV. Specifically, among individuals with low ETV there should be a significantly stronger correlation, while among individuals with high ETV there should be a significantly weaker correlation.

Currently the authors do not report such a test. They do show that ETV is unrelated to "learned harmfulness", while it is negatively related to both "trait information" and "trust behavior". But the most interesting theoretical interpretation of this result—and the possibility that the authors foreground in the discussion—depends upon the claim that in healthy/normal individuals there is a linkage between "learned harmfulness" and "trait"/"behavior", and that this linkage is broken by ETV. The current analytic approach provides circumstantial rather than direct statistical evidence for this claim, and as far as I can see it would be easy for the authors to conduct the more direct test.

2. The authors note that "the present sample is limited to male offenders, thus it is unclear whether or how gender may impact [the results]". I think it is important for the authors to be much more explicit in noting another, and arguably even larger, consequence of testing only "male offenders": It is unclear whether or how being an incarcerated offender may impact the results. This study is motivated entirely from the perspective of "identifying and specifying the way in which learning is disrupted and can affect behavior in individuals exposed to violence" (56-57). The introduction does not mention incarceration or criminal defense except in the context of noting the current methods, and the discussion briefly touches on it in the most general terms. Yet, obviously many—probably most—individuals exposed to violence are not incarcerated offenders. And, plausibly, the experience of being an incarcerated offender could play a large role in shaping behavior on these tasks, including the role of ETV as a moderator. If the authors are going to frame their manuscript in terms of its implications for the general population of people exposed to violence, then they need to be much more explicit about the limitation introduced by testing these implications solely in incarcerated offenders. (Although not necessarily crucial, it would be worth at least considering replicating the experiment in non-incarcerated individuals with variable exposures to violence).

Minor points:

1. 220-221: "Accurately" appears twice

2. I didn't understand exactly why the authors were standardizing variables before adding them into regression models in some analyses, and I wonder if they could address this. (Although this would change the coefficients estimated in the relevant models I don't see any reason it would change

significance levels, so I'm not concerned from that perspective). Additionally, I felt that Figure 3 would have been more informative with non-standardized effects plot. Although I am not certain, I am worried that the standardization procedure could at least possibly have inflated the visual impression that the slopes were of near-identical magnitude for bad and good agents.

Reviewer #2 (Remarks to the Author):

This is a well written and clear paper. The topic is interesting and methods and analysis appropriate. Examining prisoners with different degrees of past exposure to violence the authors conclude that "Exposure to community violence affects the development of subjective moral impressions and trust behavior". My main concern is the ability of the authors to make general claims about community violence when examining only a very particular group that have been exposed to such violence – individuals who have committed serious crime and are now serving time in a high security prisons. It is extremely likely that those individuals differ in many ways from non-prisoners who have been exposed to violent crime. For example, this group may be different from non-prisoners in moral attitudes, risk taking, personal relationships, social abilities, risk taking, perhaps even hormonal levels. It is impossible to know whether exposure to violence indeed impacts upon trust behavior in general or if this is only true in people who end up committing crime. There are two solutions to this problem (i) run a large sample of non-prisoners (ii) re-write the study constricting the research question and conclusions to prisoners. I believe the first option is preferred as this is the one that would produce a paper of general interest.

Other comments:

1. The authors collect responses from several questionnaires. Did they examine correlations between task behavior and other questionnaires apart from ECV? If so multiple corrections need to be applied.
2. One may be concerned that high ECV subjects are simply paying less attention thus not differentiating between agents. However, the fact that accuracy rate does not differ with ECT suggests otherwise. I would make this point in the paper.
3. I could not find the reference for Siegel et al (in press) in the reference list
4. Pg 9 – the authors say that less discrimination between agents in the task was related to violent acts inside the prison. However, the reason behind this result could be that high ECT is related to violent acts in the prison and agent discrimination is correlated to ECT? If ECT is controlled for I would imagine the relationship would go away?

Response to Reviewer #1

We are thankful that this Reviewer thought that the research was “interesting”, and the methods were “sound.” We believe we addressed this Reviewer’s concerns (see below) and now hope the paper is ready for publication.

1. The main theoretical conclusion is that “exposure to violence does not fundamentally disrupt learning processes, but instead may produce a problem with generating global social impressions based on learned information and translating these impressions into adaptive decision-making.” (367-370) The most direct statistical test of this hypothesis would be to show that the correlation between “learned harmfulness” and “trait information and/or trust behavior” is moderated by ETV. Specifically, among individuals with low ETV there should be a significantly stronger correlation, while among individuals with high ETV there should be a significantly weaker correlation.

Currently the authors do not report such a test. They do show that ETV is unrelated to “learned harmfulness”, while it is negatively related to both “trait information” and “trust behavior”. But the most interesting theoretical interpretation of this result—and the possibility that the authors foreground in the discussion—depends upon the claim that in healthy/normal individuals there is a linkage between “learned harmfulness” and “trait”/“behavior”, and that this linkage is broken by ETV. The current analytic approach provides circumstantial rather than direct statistical evidence for this claim, and as far as I can see it would be easy for the authors to conduct the more direct test.

We appreciate this Reviewer’s point and analysis suggestion. There are three parts that need to be addressed: the conceptual relationship between objective and subjective learning, the learning-ETV-trust analysis, and the sentence the Reviewer points out.

First, it is important to note that we have two measures of “learned harmfulness” using the Siegel et al. (2018) task. We can measure *objective learning* about the agent’s harm preferences, and also how this learned information is translated into *subjective impressions* of the agents. Previous research indicates that objective learning and subjective impressions are not necessarily related^{1,24}. Participants rapidly form subjective impressions about moral character from very little information (e.g., after only a few trials – see Fig. 2a in Siegel et al. 2018). However, objective learning of harmfulness requires integrating over more information and updated beliefs gradually over a longer timescale, reflecting the fact that harmfulness represents the precise *exchange rate between money and pain*, which cannot be inferred from a single trial. These two processes are not necessarily directly related, because the accuracy of objective learning can be quantified, whereas subjective impressions by definition do not have a “correct” answer: one person might judge an agent who requires \$1 per shock to inflict pain on a stranger is highly immoral and untrustworthy, whereas another person (who themselves would only require \$0.10 per shock) might view the same agent as extraordinarily generous. Thus, two participants whose objective learning is identical (e.g., they have both reached 100% correct predictions by the end of the task) may have very different subjective impressions and subsequent trust behavior. In the manuscript, we added text to address the conceptual difference between objective and subjective learning (see **page 2**), as well as, the replication of distinct effects with these measures and behavior (see **pages 7-10**).

***Page 2:** Research on social learning has shown that there are two distinct components of harmfulness learning. On the one hand, people use social cues to objectively update*

their beliefs about others' harmfulness by gradually accumulating information over time to predict future outcomes (i.e., in a Bayesian manner¹⁹). On the other hand, people form subjective impressions about moral character that emerge rapidly and effortlessly^{20,21}. These beliefs and moral impressions are used to adaptively learn and decide whom to trust in social interactions^{17,22}. For example, in a study by Siegel and colleagues¹⁹, participants entrusted more money to agents who were less willing to harm others for profit and ascribed better moral character (subjective impression) to those agents compared to those agents who were more willing to harm for profit and had worse moral character. Together, these components of learning about other's harmfulness serve as powerful informational tools; for the purpose of survival, humans are evolutionarily inclined to identify potential foes and avoid them through adaptive social decision-making^{23,24}.

Second, the reviewer suggests that we perform a moderation analysis testing whether the correlation between learned harmfulness and trust behavior is moderated by ETV scores. In our original paper on **page 10**, we reported the trust behavior mediation analyses including subjective impressions and ETV. We opted for a mediation analysis as we predicted subjective impressions would serve as the mechanistic link between ETV scores and trust behavior. Additionally, our design afforded temporal order that supports a mediation analysis. However, in our original submission we did not provide analyses connecting objective learning, ETV and trust because there was no relationship between objective learning and exposure to violence (see **page 7**). That being said, we certainly take the Reviewer's point and now address it here.

Our two measures of objective learning were (1) how well participants' predicted the agents' choices, i.e., prediction accuracy, and (2) the model estimate of belief volatility, ω , which captures the rate at which beliefs evolve over time. We had no *a priori* reason to believe that these metrics describing how information was integrated would impact subsequent trust behavior (referring to model estimate ω) after learning was complete. However, one could argue that a person who is less able to predict the agents' choices would have greater difficulty distinguishing in their subjective impressions and trust behavior for agents who behave differently. When we examine this association, we do see, as the Reviewer suggests, that in general, objective learning accuracy and trust are linked ($P < 0.001$), where increased accuracy was associated with a greater tendency to adapt trust behavior to agents with different harm preferences. However, that association is not moderated by ETV score ($\beta = -0.294 \pm 12.960$, $t = 0.0227$, $P = 0.982$) (nor mediated, indirect effect of accuracy \rightarrow ETV score \rightarrow trust 2.181 ± 14.753 , CI [-28.412, 31.627]). Together, **these findings suggest that exposure to violence does not impact the association between the ability to learn preferences of others, and moreover, use that information to engage in trust behavior. However, as demonstrated in the primary analyses, exposure to violence does impact the ability to form subjective impressions based on distinguishable behaviors, and subsequently adapt trust behavior accordingly.** We now include this information as a footnote in the manuscript (see **page 10**).

Page 10: *It is possible that objective learning and trust are associated, such that a person who is less able to predict the agents' choices would have greater difficulty distinguishing trust behavior for agents who behave differently. In fact, we find a significant association between accuracy and trust behavior ($P < 0.001$), where increased accuracy was associated with a greater tendency to adapt trust behavior to agents with different harm preferences. However, there was no impact of ETV score on that relationship. Together, these findings suggest that exposure to violence does not impact the association between the ability to learn preferences of others, and moreover, use that information to engage in trust behavior. However, as demonstrated in the main*

analysis, exposure to violence does impact the ability to form subjective impressions based on distinguishable behaviors, and subsequently adapt trust behavior accordingly.

Finally, to address the concerns about the cited sentence, we made the following modification (see **page 12**). It is worth noting that the content of this sentence reflects the deficiencies in the subjective aspects of harmfulness learning associated with exposure to violence, and the absence of any effects with objective learning.

Page 12: *On the whole, these findings raise the intriguing possibility that exposure to violence does not fundamentally disrupt all components of social learning, but instead may produce a problem with generating global subjective social impressions and translating those impressions into adaptive social decision-making.*

2. The authors note that “the present sample is limited to male offenders, thus it is unclear whether or how gender may impact [the results]”. I think it is important for the authors to be much more explicit it noting another, and arguably even larger, consequence of testing only “male offenders”: It is unclear whether or how being an incarcerated offender may impact the results. This study is motivated entirely from the perspective of “identifying and specifying the way in which learning is disrupted and can affect behavior in individuals exposed to violence” (56-57). The introduction does not mention incarceration or criminal defense except in the context of noting the current methods, and the discussion briefly touches on it in the most general terms. Yet, obviously many—probably most—individuals exposed to violence are not incarcerated offenders. And, plausibly, the experience of being an incarcerated offender could play a large role in shaping behavior on these tasks, including the role of ETV as a moderator. If the authors are going to frame their manuscript in terms of its implications for the general population of people exposed to violence, then they need to be much more explicit about the limitation introduced by testing these implications solely in incarcerated offenders. (Although not necessarily crucial, it would be worth at least considering replicating the experiment in non-incarcerated individuals with variable exposures to violence).

Please see our response to the Editor’s first comment above for the scientific rationale behind choosing to test our hypotheses in an incarcerated sample, and the changes we made to the revised manuscript to address these points.

Minor points

1. 220-221: “Accurately” appears twice

Thank you for pointing out this error. We corrected this sentence in the revised manuscript.

2. I didn’t understand exactly why the authors were standardizing variables before adding them into regression models in some analyses, and I wonder if they could address this. (Although this would change the coefficients estimated in the relevant models I don’t see any reason it would change significance levels, so I’m not concerned from that perspective). Additionally, I felt that Figure 3 would have been more informative with non-standardized effects plot. Although I am not certain, I am worried that the standardization procedure could at least possibly have inflated the visual impression that the slopes were of near-identical magnitude for bad and good agents.

We chose to standardize the variables before adding them in to the regression because that is the recommended approach when including interactions in regression models³⁰. That being said, we recognize the Reviewer’s point regarding the possible visual inflation of the slopes. Below, we plotted Figure 3 with both the unstandardized (top) and standardized (bottom) variables. As you can see, the main difference between the unstandardized and standardized figures comes from differences in scaling driven by the range of the y-axis. Because the unstandardized figures result in a larger range in the y-axis, the differences between agents may appear smaller (simply because they are less ‘zoomed in’). Based on the statistical rationale and consistency between how the models were conducted and visually presented, we opted to keep the standardized figures. However, if the Editor or Reviewer prefers, we could add the unstandardized figures to the Supporting Information.

Figure 3 unstandardized:

Figure 3 standardized:

Response to Reviewer #2

We also are very thankful that this Reviewer found our manuscript to be “well written” that covers a “interesting” question with “appropriate” methods. We tried to address all of this Reviewer’s outstanding suggestions, thereby making the manuscript even stronger.

1. Examining prisoners with different degrees of past exposure to violence the authors conclude that “Exposure to community violence affects the development of subjective moral impressions and trust behavior”. My main concern is the ability of the authors to make general claims about community violence when examining only a very particular group that have been exposed to such violence – individuals who have committed serious crime and are now serving time in a high security prisons. It is extremely likely that those individuals differ in many ways from non-prisoners who have been exposed to

violent crime. For example, this group may be different from non-prisoners in moral attitudes, risk taking, personal relationships, social abilities, risk taking, perhaps even hormonal levels. It is impossible to know whether exposure to violence indeed impacts upon trust behavior in general or if this is only true in people who end up committing crime. There are two solutions to this problem (i) run a large sample of non-prisoners (ii) re-write the study constricting the research question and conclusions to prisoners. I believe the first option is preferred as this is the one that would produce a paper of general interest.

Please see our response to the Editor's first comment above for the scientific rationale behind choosing to test our hypotheses in an incarcerated sample, and the changes we made to the revised manuscript to address these points. Moreover, from a representativeness of offender perspective, it is important to note that not all individuals in a high-security prison are extreme examples of offenders. In the United States, assignment to a high security prison is affected by a lot of factors outside of crime alone (e.g., space, sentence length, availability of required programming, repeat offender status even if for low-level drug crime).

2. The authors collect responses from several questionnaires. Did they examine correlations between task behavior and other questionnaires apart from ECV? If so multiple corrections need to be applied.

We set out to test the hypotheses outlined in the current manuscript (see **page 3**). In addition, we collected a battery of other questionnaires to control for other factors commonly found in incarcerated samples and that tend to be related to ETV. The inclusion of these variables was to address potential confounding factors and examine the specificity of the ETV-task relationships. There was no intent to examine the relationships between those other factors and task performance (i.e., we had no hypotheses related to these variables and task performance). We clarified this on **pages 3** and **5**. Typically, it is not recommended to apply multiple corrections when several regressions including a potential confound are performed because we tend to vary and adjust existing models rather than test different questions³¹. For the sake of completeness, in the tables presented in Supporting Information we provide the results when including these potential confounding variables (see **Tables S4-6** in **Supporting Information**). That being said, to fully address the Reviewer's point, we did re-run the primary analyses applying multiple corrections. All of the primary effects (i.e., interactions between ETV and agent on subjective ratings and trust behavior) remain significant. The adjusted p-values following Bonferroni correction, correcting for 3 questionnaires (Hare Psychopathy Checklist Revised, Antisocial Personality Disorder, and Childhood Trauma Questionnaire) were: Subjective impression rating, $P < 0.001$; subjective uncertainty rating, $P < 0.001$; trust behavior, $P = 0.024$. If the Editor and Reviewer thinks it is necessary, we can add a footnote about these effects surviving Bonferroni correction.

3. One may be concerned that high ECV subjects are simply paying less attention thus not differentiating between agents. However, the fact that accuracy rate does not differ with ECT suggests otherwise. I would make this point in the paper.

Thank you for suggesting that we highlight this important point, signifying that aberrant impressions and trust behavior in participants with high ETV scores is unlikely to result from differences in attention or motivation. We highlighted this point on **page 7** of the revised manuscript:

Page 7: *There was no relationship between ETV score and prediction accuracy for either agent (Spearman's ρ , good: $\rho = -0.065$, $p = .483$; bad: $\rho = 0.043$, $p = .639$). This suggests that participants with higher exposure to violence were equally motivated to learn the harm preferences of the agents, relative to those with lower exposure to violence.*

We also reference the result on **page 11** of the Discussion.

4. I could not find the reference for Siegel et al (in press) in the reference list

We apologize for that oversight. The reference for Siegel et al., (in press) has been added to the reference list in the revised manuscript. We also included the reference at the end of this letter¹ (Siegel et al., 2018).

5. Pg 9 – the authors say that less discrimination between agents in the task was related to violent acts inside the prison. However, the reason behind this result could be that high ECT is related to violent acts in the prison and agent discrimination is correlated to ECT? If ECT is controlled for I would imagine the relationship would go away?

Given the strong association between exposure to violence and violent behavior, we agree that violent acts in the prison also are likely to correlate with ETV scores. And thus, we agree it is likely that if ETV scores are controlled for, the relationship between agent discrimination in trust behavior and the number of prison violations would go away. In fact, we do find that higher ETV scores predict more violent acts in prison ($\rho = 0.450$, $p < .001$), and that the relationship between adaptive trust and violent acts in prison is not significant when ETV scores are controlled for ($\rho = -0.087$, $p = .347$). However, because we predict that the relationship between adaptive trust and prison violations is linked to exposure to violence, we do not believe that controlling for ETV scores is appropriate in this instance. We apologize for not making this clearer in the original manuscript and edited the relevant text to make the predicted associations clearer (see **page 10**).

This comment, however, **inspired us to perform additional analyses to assess whether the relationship between ETV score and violent acts in prison is mediated by the extent to which one adapts trust behavior towards agents with varying harm preferences (Δ trust), as a function of impression sensitivity.** A serial multiple mediation analysis was used to investigate the hypothesis that impression sensitivity and adaptive trust mediate the effect of ETV scores on the number of violations in prison. Results indicated that ETV was a significant predictor of impression sensitivity, as measured by Δ judgment, $b = -0.025$, $SEM = 0.012$, $p = 0.041$, however impression sensitivity was not an independent predictor of violations in prison, $b = 4.597$, $SEM = 2.735$, $p = 0.096$. We found that ETV was also a significant predictor of adaptive trust behavior (Δ invest), $b = -2.341$, $SEM = 0.863$, $p = .008$, and that adaptive trust behavior was a significant predictor of violations in prison, $b = -0.121$, $SEM = 0.035$, $p = .001$. ETV score was only a marginally significant predictor of prison violations after impression sensitivity and trust behavior were accounted for (effect = 0.622 ± 0.340 , $p = 0.070$). The indirect effects were tested using a bootstrap estimation approach with 5000 samples. These results indicated the indirect serial coefficient was significant, effect = 0.099 ± 0.071 , 95% CI = [0.002, 0.274], **suggesting that disruptions in the ability to form distinguishable impressions resulting from higher ETV scores, translates into maladaptive trust behavior, which in turn leads to an increased frequency of direct violations in prison.**

This analysis has been added to **page 10** of the revised manuscript:

Page 10: Indeed, higher ETV scores predict more behavioral violations in prison ($\rho = 0.450, p < .001$). However, we predicted that this relationship would be mediated by the extent to which participants differentiated in their subjective impressions and trust behavior between the good and bad agent. Consequently, we applied a serial multiple mediation analysis using the PROCESS macros for SPSS³² (model 6) that allowed us to determine the causal link between mediators with a specified direction of causal flow. We investigated whether the relationship between exposure to violence and prison violations was mediated by trust behavior (Δ trust) as a function of impression sensitivity (Δ judgment). ETV score was only a marginally significant predictor of prison violations after impression sensitivity and trust behavior were accounted for (effect = $0.622 \pm 0.340, p = 0.070$). The indirect effects were tested using a bootstrap estimation approach with 5000 samples. These results indicated the indirect serial coefficient was significant, effect = $0.099 \pm 0.071, 95\% CI = [0.002, 0.274]$ (see **Supporting Information** for full mediation results and **Fig. S1**), suggesting that disruptions in the ability to form distinguishable impressions resulting from higher ETV scores, translates into maladaptive trust behavior, which in turn leads to an increased frequency of direct violations in prison.

We added the full mediation results to the Supporting Information, **page 3**:

Impression sensitivity and maladaptive trust mediate the relationship between ETV and prison violations. Serial multiple mediation analysis was used to investigate the hypothesis that the extent to which one differentiates in subjective impressions and adapts trust behavior towards agents with varying harm preferences mediates the effect of ETV on the number of violations in prison. Results indicated that ETV was a significant predictor of impression sensitivity (Δ judgment), $b = -0.025, SEM = 0.012, p = 0.041$, however impression sensitivity was not an independent predictor of violations in prison, $b = 4.597, SEM = 2.735, p = 0.096$. We found that ETV was also a significant predictor of adaptive trust behavior (Δ invest), $b = -2.341, SEM = 0.863, p = .008$, and that adaptive trust behavior was a significant predictor of violations in prison, $b = -0.121, SEM = 0.035, p = .001$. ETV was only a marginally significant predictor of prison violations after controlling for the mediators, $b = 0.622, SEM = 0.340, p = 0.070$. When considering the mediating variables separately and together in relation to the mediating indirect effects of ETV on the number of prison violations, single mediation of Δ invest was significant ($b = 0.284, SEM = 0.165, 95\% CI = 0.035, 0.660$), and the serial-multiple mediation of Δ judgment and Δ invest was significant ($b = 0.099, SEM = 0.072, 95\% CI = 0.002, 0.274$). The single mediation of Δ judgment was not statistically significant ($b = -0.115, SEM = 0.078, 95\% CI = -0.296, 0.006$).

Figure S1. Serial Multiple Mediation Analysis. n.s = not significant; *P < 0.05; **P < 0.01; ***P < 0.001

References

1. Siegel, J. Z., Mathys, C., Rutledge, R. B. & Crockett, M. J. Beliefs about bad people are volatile. *Nat. Hum. Behav.* **2**, 750 (2018).
2. Casciano, R. & Massey, D. S. Neighborhoods, employment, and welfare use: Assessing the influence of neighborhood socioeconomic composition. *Soc. Sci. Res.* **37**, 544–558 (2008).
3. Garbarino, J. & Sherman, D. High-Risk Neighborhoods and High-Risk Families: The Human Ecology of Child Maltreatment. *Child Dev.* **51**, 188–198 (1980).
4. Goffman, A. *On the Run: Fugitive Life in an American City*. (Picador, 2015).
5. Monahan, K. C., King, K. M., Shulman, E. P., Cauffman, E. & Chassin, L. The effects of violence exposure on the development of impulse control and future orientation across adolescence and early adulthood: Time-specific and generalized effects in a sample of juvenile offenders. *Dev. Psychopathol.* **27**, 1267–1283 (2015).
6. Tangney, J. P., Stuewig, J. & Mashek, D. J. Moral Emotions and Moral Behavior. *Annu. Rev. Psychol.* **58**, 345–372 (2007).
7. Clear, T. R., Rose, D. R. & Ryder, J. A. Incarceration and the Community: The Problem of Removing and Returning Offenders. *Crime Delinquency* **47**, 335–351 (2001).
8. Baskin, D. & Sommers, I. Trajectories of Exposure to Community Violence and Mental Health Symptoms Among Serious Adolescent Offenders. *Crim. Justice Behav.* **42**, 587–609 (2015).
9. Fowler, P. J., Tompsett, C. J., Braciszewski, J. M., Jacques-Tiura, A. J. & Baltes, B. B. Community violence: A meta-analysis on the effect of exposure and mental health outcomes of children and adolescents. *Dev. Psychopathol.* **21**, 227–259 (2009).
10. Javdani, S., Abdul-Adil, J., Suarez, L., Nichols, S. R. & Farmer, A. D. Gender Differences in the Effects of Community Violence on Mental Health Outcomes in a Sample of Low-Income Youth Receiving Psychiatric Care. *Am. J. Community Psychol.* **53**, 235–248 (2014).
11. DuRant, R. H., Pendergrast, R. A. & Cadenhead, C. Exposure to violence and victimization and fighting behavior by urban black adolescents. *J. Adolesc. Health* **15**, 311–318 (1994).
12. Guerra, N. G., Huesmann, L. R. & Spindler, A. Community Violence Exposure, Social Cognition, and Aggression Among Urban Elementary School Children. *Child Dev.* **74**, 1561–1576 (2003).
13. Hawkins, J. D. *et al.* Predictors of Youth Violence. *Juvenile Justice Bulletin*. (2000).
14. Anderson, E. The Code of the Streets. *Monthly Atlantic* 81–94 (1994).
15. Besbris, M., Faber, J. W., Rich, P. & Sharkey, P. Effect of neighborhood stigma on economic transactions. *Proc. Natl. Acad. Sci.* 201414139 (2015).
doi:10.1073/pnas.1414139112
16. Raudenbush, D. “I Stay by Myself”: Social Support, Distrust, and Selective Solidarity Among the Urban Poor. (2016).
17. Wilson, W. J. *More Than Just Race: Being Black and Poor in the Inner City (Issues of Our Time)*. (W. W. Norton & Company, 2009).
18. Buka, S. L., Stichick, T. L., Birdthistle, I. & Earls, F. J. Youth Exposure to Violence: Prevalence, Risks, and Consequences. *Am. J. Orthopsychiatry* **71**, 298–310 (2001).
19. Hetey, R. C. & Eberhardt, J. L. The Numbers Don’t Speak for Themselves: Racial Disparities and the Persistence of Inequality in the Criminal Justice System
The Numbers Don’t Speak for Themselves: Racial Disparities and the Persistence of Inequality in the Criminal Justice System. *Curr. Dir. Psychol. Sci.* **27**, 183–187 (2018).
20. Fehr, E. On the Economics and Biology of Trust. *J. Eur. Econ. Assoc.* **7**, 235–266 (2009).
21. Knack, S. & Keefer, P. Does Social Capital Have an Economic Payoff? A Cross-Country Investigation. *Q. J. Econ.* **112**, 1251–1288 (1997).

22. LaPorta, R., Lopez-de-Silanes, F., Shleifer, A. & Vishny, R. W. *Trust in Large Organizations*. (National Bureau of Economic Research, 1996). doi:10.3386/w5864
23. Moffitt, T. E. Childhood exposure to violence and lifelong health: Clinical intervention science and stress biology research join forces. *Dev. Psychopathol.* **25**, (2013).
24. Todorov, A., Pakrashi, M. & Oosterhof, N. N. Evaluating Faces on Trustworthiness After Minimal Time Exposure. *Soc. Cogn.* **27**, 813–833 (2009).
25. Engell, A. D., Haxby, J. V. & Todorov, A. Implicit Trustworthiness Decisions: Automatic Coding of Face Properties in the Human Amygdala. *J. Cogn. Neurosci.* **19**, 1508–1519 (2007).
26. Haidt, J. & Joseph, C. Intuitive ethics: how innately prepared intuitions generate culturally variable virtues. *Daedalus* **133**, 55–66 (2004).
27. Stanley, D. A., Sokol-Hessner, P., Banaji, M. R. & Phelps, E. A. Implicit race attitudes predict trustworthiness judgments and economic trust decisions. *Proc. Natl. Acad. Sci.* **108**, 7710–7715 (2011).
28. Alexander, R. *The Biology of Moral Systems (Foundations of Human Behavior)*. (Aldine Transaction, 1987).
29. Gintis, H. Strong reciprocity and human sociality. *J. Theor. Biol.* **206**, 169–179 (2000).
30. Aiken, L. S., West, S. G. & Reno, R. R. *Multiple Regression: Testing and Interpreting Interactions*. (SAGE, 1991).
31. Fiedler, K., Kutzner, F. & Krueger, J. I. The Long Way From α -Error Control to Validity Proper: Problems With a Short-Sighted False-Positive Debate. *Perspect. Psychol. Sci.* **7**, 661–669 (2012).
32. Hayes, A. F. PROCESS: A versatile computational tool for observed variable mediation, moderation, and conditional process modeling. (2012).

Reviewers' comments:

Reviewer #1 (Remarks to the Author):

In my initial review I outlined two main concerns. The reviewers give careful attention to each.

The first was that the main theoretical conclusion as stated in the original manuscript suggested a key statistical test (a moderation analysis; i.e, an interaction effect) that had not been conducted. Specifically, the original article concluded that "exposure to violence ... may produce a problem with generating global social impressions based on [appropriately] learned information...". In other words, the argument was that "objective" learning ordinarily supports "subjective" impressions and thus trust behavior, but high ETV breaks the first link in this path.

In their response the authors performed the relevant test and find that it does not support this conclusion. As a result the revised manuscript emphasizes a related (and interesting) conclusion, but one that differs. It claims a dissociation between "objective" learning (which is preserved) and "subjective" learning (which is disrupted, with downstream effects on trust behavior as demonstrated in their mediation analysis). This revised claim is consistent with the evidence and statistical analyses.

However, there remain two key places where the authors overlooked the necessary update from the phrasing of the old claim to the phrasing of the new claim. In each of these places the authors argue that the specific effect of ETV is to disrupt the "translation" of objective to subjective information, but the null findings of the moderation analysis would seem to point against this conclusion, as the authors note in their reply to the reviewers.

1. Abstract: "However, exposure disrupted the ability to translate those predictions into moral impressions that dissociated between agents."
2. General discussion: "The present study identifies a specific deficit in the ability of individuals exposed to violence to translate learned social information into adaptive social behavior."
3. General discussion: "However, exposure to violence appeared to disrupt the translation of that objective encoding into subjective, global impressions of other's moral character."

My second concern was whether the exclusive use of incarcerated individuals could support a manuscript that was framed as a general test of the relationship between ETV and trait inference / trust, given the ways in which incarcerated individuals likely differ from the general population. The authors responded to this concern in four basic ways.

Two of these presented reasons in favor of studying incarcerated individuals:

1. The use of incarcerated individuals was important because it allowed for a sufficiently large sample of people with high ETV. (I.e., this was the most practical/efficient sampling method).
2. It is important to directly study incarcerated individuals because there are many of them and they are of general social importance.

If the use of incarcerated individuals is just about convenience, so be it. If, however, it is important to study incarcerated individuals because they are a distinctive population with distinctive features and must be understood in their own right, then this paper ought to be framed as a study of incarcerated

individuals and not as a general study of ETV that just happens to use incarcerated individuals. To the extent that this population is special and requires focused empirical investigation, then it is probably not the right sample from which to draw very general conclusions.

The other two points the authors brought up were not reasons to study incarcerated individuals but, rather, reasons not to worry about doing so.

3. Incarcerated individuals are generally not that different from non-incarcerated individuals, especially those who come from similar communities.

4. Incarcerated individuals show the same basic main effects as a non-incarcerated sample used in a prior study involving the same task, and there are no effects of years in prison.

3 was not convincing. The strongest claims were that incarcerated individuals do not differ substantially from others in communities "of concentrated disadvantage" and, at times, "enriched for antisocial behavior". Even if incarcerated individuals are similar to non-incarcerated individuals from these communities, are these representative communities from which to draw general psychological claims about exposure to violence? I do not have expertise in this area and so I cannot say, but it sounds as if these may be relatively atypical rather than typical communities. Also, at some very basic level, surely being incarcerated is general impactful? I will say again that I lack expertise in this area, however, so maybe my impressions on this point are leading me astray.

4 was much more on point. I don't think that these analyses are decisive in showing that whatever we learn about the ETV/trait inference/trust link among incarcerated males generalizes to the population, but they are certainly relevant.

As a result I suggest that the authors de-emphasize points 2 and 3, while emphasizing points 1 and 4. In other words, the basic framing should be that the authors studied incarcerated individuals for reasons of research practicality and efficiency, hoping that they would not differ from non-incarcerated individuals (and, thus, that there would be nothing particularly distinctive or important about their status as non-incarcerated individuals). And, as far as the authors can tell, the (mixed-ETV) population of incarcerated individuals looks very similar to the (low-ETV) population they previously tested, and there were no observed effects of years of incarceration.

If this is the framing the authors will adopt, however, I do think that the potential limitations in generalizing this result ought to be foregrounded a bit more in the introduction and the general discussion.

Reviewer #2 (Remarks to the Author):

My major concern was that the paper makes general claims about the impact of exposure to community violence on trust behavior based on data from a specialized population – individuals in a high security prison. These individuals are different in two important ways: (1) they are currently living in a very different environment than free citizens, (2) they have committed serious crimes. Living in an environment with limited agency and being separated from family and loved ones will impact subjects in fundamental ways, even if when they are freed they will no longer show differences in cognition, emotion and social interaction (although - I find this last point unlikely – being in prison has long-term negative impact on individuals' mental health). I am unconvinced that such negative impact need be correlated with time in prison – that is the negative magnitude of being in prison could

potentially be similar across different lengths of time served (or the relationship can be non-linear).

Thus, while I agree studying prisoners is important, it is still the case that one cannot make general conclusions about moral attitudes and behaviour from studying a group of subjects living in prison. The paper, including the title ("Exposure to community violence affects the development of subjective moral impressions and trust behavior"), part of the abstract and the conclusions still make general claims that are not supported by the data. As an example I cite the first paragraph of the conclusions, in which there is no mention to the specialized sample:

"The ability to infer other's intentions and predict their behavior is crucial for successful social interactions. In particular, learning whether others are likely to harm us is important for consequential social decisions like deciding whom to trust. The current data suggest that exposure to violence adversely impacts some components of harm learning, but not all. Individuals with higher ETV scores showed an ability to develop accurate beliefs about others by objectively encoding their harm preferences. However, exposure to violence appeared to disrupt the translation of that objective encoding into subjective, global impressions of other's moral character. Individuals with higher ETV scores formed more positive and less uncertain impressions of harmful agents and more negative and less certain impressions of helpful agents. Moreover, these differences in subjective impressions associated with higher ETV scores led to maladaptive trust behavior, such that individuals with higher ETV scores extended less trust than optimal when interacting with a "good" agent. Finally, the link between exposure to violence and maladaptive trusting behavior was mediated by the disturbances in impression formation. On the whole, these findings raise the intriguing possibility that exposure to violence does not fundamentally disrupt all components of social learning, but instead may produce a problem with generating global subjective social impressions and translating those impressions into adaptive social decision-making."

A general claim cannot be made by studying a population living in an environment that is fundamentally different. And so my initial recommendation is the same – either conduct a replication study on non-prisoners or write a paper about prisoners.

Response to Reviewer #1

1. ...There remain two key places where the authors overlooked the necessary update from the phrasing of the old claim to the phrasing of the new claim. In each of these places the authors argue that the specific effect of ETV is to disrupt the “translation” of objective to subjective information, but the null findings of the moderation analysis would seem to point against this conclusion, as the authors note in their reply to the reviewers.

We appreciate the Reviewer’s thorough read of the manuscript. We carefully reviewed each and every sentence in this revision for accuracy and completeness. Below we provide the specific adjustments made to the sentences this Reviewer highlighted.

- a. **Abstract: “However, exposure disrupted the ability to translate those predictions into moral impressions that dissociated between agents.”**

However, exposure to violence disrupted the ability to form moral impressions that dissociated between agents with distinguishable harm preferences.

- b. **General discussion: “The present study identifies a specific deficit in the ability of individuals exposed to violence to translate learned social information into adaptive social behavior.”**

The present study identifies a specific deficit in the ability of incarcerated individuals exposed to violence to adapt social behavior towards agents with distinguishable harm preferences.

- c. **General discussion: “However, exposure to violence appeared to disrupt the translation of that objective encoding into subjective, global impressions of other’s moral character.”**

However, exposure to violence appeared to disrupt the formation of subjective, global impressions of other’s moral character from observed harm behavior.

2. **My second concern was whether the exclusive use of incarcerated individuals could support a manuscript that was framed as a general test of the relationship between ETV and trait inference / trust, given the ways in which incarcerated individuals likely differ from the general population. If the use of incarcerated individuals is just about convenience, so be it. If, however, it is important to study incarcerated individuals because they are a distinctive population with distinctive features and must be understood in their own right, then this paper ought to be framed as a study of incarcerated individuals and not as a general study of ETV that just happens to use incarcerated individuals. To the extent that this population is special and requires focused empirical investigation, then it is probably not the right sample from which to draw very general conclusions. As a result I suggest that the authors de-emphasize points 2 and 3, while emphasizing points 1 and 4. In other words, the basic framing should be that the authors studied incarcerated individuals for reasons of research practicality and efficiency, hoping that they would not differ from non-incarcerated individuals (and, thus, that there would be nothing particularly distinctive or important about their status as non-incarcerated individuals). And, as far as the authors can tell,**

the (mixed-ETV) population of incarcerated individuals looks very similar to the (low-ETV) population they previously tested, and there were no observed effects of years of incarceration. If this is the framing the authors will adopt, however, I do think that the potential limitations in generalizing this result ought to be foregrounded a bit more in the introduction and the general discussion.

We very much appreciate the Reviewer's thoughts on the arguments laid out in our last response and revised manuscript. We apologize for the lack of clarity regarding our decision to use incarcerated individuals to study the exposure to violence on harmfulness learning (Reviewer's noted point 1). We did not intend to imply that our decision was made merely for convenience or because they are a distinctive population with distinct features. Rather, the decision was made to maximize variation in our main independent variable, exposure to violence (ETV). We strategically selected this purposive sampling method because prior work in the senior author's lab suggested that it would not only be scientifically unwarranted, but also resource prohibitive to test our primary research questions using probability sampling methods. Although we successfully have run our harmfulness learning task in online samples previously (Siegel et al., 2018), it is not possible to test the hypotheses regarding exposure to violence (ETV) in the current paper in an online sample. This is because online samples do not show the full range of ETV scores observed in an incarcerated sample. Below is a histogram showing the distribution of ETV scores in a recent MTurk sample (N=591).

Exposure to Violence with MTurk Sample (n = 591)

As you can see, even in a large online sample, the range of ETV scores is restricted; only a tiny fraction of participants score at the upper range of the scale, and there are no participants who score a 12 or 13 on the scale. Thus, even if we collected an extremely large online sample (which would cost several thousand dollars, given the length of our task), we would still not observe the necessary range of ETV scores. The alternative -- collecting data in the lab with participants recruited from the local community -- is equally

unfeasible. Previous efforts of this nature run in the senior author's lab have required two full-time research assistants, more than a year of data collection, and tens of thousands of dollars. Moreover, while we are able to obtain a better range of ETV scores in these targeted community samples, they are not as well distributed as a prison sample (from the last response: the total ETV score in a sample of 387 community members enriched for antisocial behavior the ETV mean was 4.42 (SD=3.59) compared to 8.04 (SD= 3.22) for the prison sample). For these reasons, we believe that incarcerated offenders are the most scientifically appropriate sample to provide as much insight as possible into the effects of exposure to violence on harm learning.

To address the Reviewer's point, we adopted the suggested framing focusing on the methodological importance of using a sample with sufficient variability in ETV scores. Specifically, we emphasize that the decision to use an incarcerated sample was to maximize variation in exposure to violence in the introduction (**page 3**).

While a sample of currently incarcerated individuals is not the same as a sample from the general population, this type of sample does serve as an informative sample in which to explore how differences in exposure to violence impact harm learning. It is well-documented that exposure to violence among the incarcerated covers the full continuum of potential experiences compared to the general population where scores are often restricted in range and narrowly centered around a few points within that range. Moreover, by using a sample of currently incarcerated individuals, we are better poised to investigate the variation in exposure to violence within a sample that is already demonstrating the theorized behavioral effects of such exposure.

Additionally, we more clearly emphasized that our population of incarcerated individuals show the same basic learning effects that we previously observed in non-incarcerated samples (Siegel et al. 2018; Reviewer point 4). We discuss the potential limitations in generalizing the results due to the current sample.

*Before concluding, methodological and conceptual limitations should be noted. The present sample is limited to incarcerated offenders, thus we do not know whether or how incarceration-status may impact the relationship between exposure to violence and harm learning. However, it is important to note that all task main effects replicated previous research in non-incarcerated samples. For instance, previous work using the same task has shown that people form less positive, more uncertain, and more volatile beliefs about the bad agent, relative to the good agent, and adjust their trust behavior according to the harm preferences of the agents¹. We observe the same pattern of results in our sample of incarcerated individuals. Moreover, length of incarceration (see **Supporting Information**) and other correlates known to increase risk for incarceration did not impact the reported exposure to violence effects. Ultimately, being currently incarcerated is just one type of adverse outcome related to exposure to violence that should not be seen as excluding the importance of the lived experience of exposure to violence for these individuals²⁻⁶.*

While it may be useful to replicate the findings in a sample of non-incarcerated individuals, this raises important experimental considerations. From a scientific perspective, using a sample with sufficient variability in ETV scores, and whose experience with exposure to violence has led to great personal cost, is essential.

Notably, the distribution of ETV scores in our sample of incarcerated individuals covers the full range of the scale. Endeavors in samples typical of the psychology research, such as University or online settings, often suffer from restricted range in ETV scores. Nonetheless, to test for generalizability, future research should replicate the present research in a sample of non-incarcerated individuals whose ETV scores are reflective of a range of experiences.

A final consideration is that implementing shocks in the harmfulness learning task is not as extreme a behavior as what might be seen in the real world (e.g., sexual assault, murder) for individuals exposed to violence or involved in the justice system. Therefore, it is possible that the objective learning of other's harm preferences could be different with more extreme behaviors. Future research should continue to investigate components of learning in those exposed to violence and vary the stimuli used to assess learning that consider cultural and situational contexts.

Finally, in light of the Reviewer's comments we toned down claims throughout the manuscript to reflect our sample. Importantly, we revised the title of the manuscript to highlight that our findings come from an incarcerated sample of male offenders (*"Exposure to community violence affects the development of subjective moral impressions and trust **behavior in a sample of incarcerated males**"*).

Therefore, in line with the Reviewer's comments, we clarified that the use of incarcerated individuals was important because it allowed for a sufficiently large sample of participants with a range of ETV scores (i.e., an important methodological consideration in any research; Reviewer point 1). We do not include any statements that mention that we have to study incarcerated individuals "in their own right" (Reviewer point 2), just that this sample is the most representative of the construct of interest for the research question. Additionally, we emphasize that "incarcerated individuals show the same basic main effects as a non-incarcerated sample used in a prior study involving the same task, and there are no effects of years in prison" (Reviewer point 4). We do not include statements saying that incarcerated individuals are no different from non-incarcerated individuals (Reviewer point 4). Rather, we focus on the impact of exposure to violence on learning and trust behavior within individuals who are incarcerated and on the larger communities in which incarcerated individuals typically experience exposure to violence. In the United States, exposure to violence is clustered in communities of disadvantage and it is well-documented that certain communities are marred by the impact of violence exposure and the rotating door of incarceration¹⁶⁻¹⁹. Given our research question is about exposure to violence and the measure is a measure of lifetime exposure, we think it is important in the discussion to extend the impact of the results to the larger communities in which many incarcerated individuals come from (e.g., ones of disadvantage). We very much appreciate the Reviewer pushing us to make clear that our question is about exposure to violence, the use of a best sample test that question, and the implications of that approach for understanding harm learning in general (which conceptually appears no different than other samples) and the link between exposure to violence and harm learning more specifically.

Response to Reviewer #2

1. My major concern was that the paper makes general claims about the impact of exposure to community violence on trust behavior based on data from a specialized population –individuals in a high security prison. These individuals are different in two important ways: (1) they are currently living in a very different environment than free citizens, (2) they have committed serious crimes. Living in an environment with limited agency and being separated from family and loved ones will impact subjects in fundamental ways, even if when they are freed they will no longer show differences in cognition, emotion and social interaction (although - I find this last point unlikely – being in prison has long-term negative impact on individuals' mental health). I am unconvinced that such negative impact need be correlated with time in prison – that is the negative magnitude of being in prison could potentially be similar across different lengths of time served (or the relationship can be non-linear).

We understand the Reviewer's concerns. There are several points that this Reviewer raises that must be addressed separately.

First, the Reviewer notes that currently incarcerated individuals are "living in a very different environment than free citizens." This is absolutely true. Being separated from family and loved ones may impact cognition and behavior in fundamental ways. This is an issue that is true for many samples, including active military, some immigrants, etc. Living in loud environments with toxins may impact cognition and behavior, as well. This is also true for samples from low-income public housing. However, the main questions that we need to address are: 1) does our sample of incarcerated individuals *within the current task* differ from samples of non-incarcerated participants, and 2) is the ETV effect robust enough to withstand consideration for variables that estimate impact and correlates of incarceration? Regarding the first question, previous work using the same task has shown that non-incarcerated individuals form less positive, more uncertain, and more volatile beliefs about the bad agent, relative to the good agent. This past work also indicates that people adjust their behavior in the trust game according to the harm preferences of the agent with whom they are interacting with. Each one of these main effects are replicated in the present sample of incarcerated individuals, suggesting that at a basic level, incarcerated and non-incarcerated participants show similar patterns of learning about good and bad agents in our task. In the revised discussion (**page 9**), we highlight these main effects more clearly.

The present sample is limited to incarcerated offenders, thus we do not know whether or how incarceration-status may impact the relationship between exposure to violence and harm learning. However, it is important to note that all task main effects replicated previous research in non-incarcerated samples. For instance, previous work using the same task has shown that people form less positive, more uncertain, and more volatile beliefs about the bad agent, relative to the good agent, and adjust their trust behavior according to the harm preferences of the agents¹. We observe the same pattern of results in our sample of incarcerated individuals.

Regarding the second question, in the manuscript we did what we could to control for general effects and we explicitly consider behavior in prison in our analysis. In response to previous comments (Reviewer 1) we investigated the effects of length of incarceration

on task behavior. We believe this is a good proxy to address the long-term effects of being incarcerated on our main effects. We also control for known correlates of incarceration, including education and antisociality. The relationships between ETV and harm learning that we observe are robust to controlling for these factors (**page 9**).

*Moreover, length of incarceration (see **Supporting Information**) and other correlates known to increase risk for incarceration did not impact the reported exposure to violence effects. Ultimately, being currently incarcerated is just one type of adverse outcome related to exposure to violence that should not be seen as excluding the importance of the lived experience of exposure to violence for these individuals²⁻⁶.*

Moreover, we directly asked whether behavior in our task was associated with real social behaviors in prison. Consistent with findings in non-incarcerated samples showing an association between ETV and aggressive/antisocial behavior⁷⁻⁹, ETV in our sample of incarcerated individuals was associated with an increased number of violations in prison. Importantly, we found that disruptions in the ability to form distinguishable impressions resulting from higher ETV scores, translated into maladaptive trust behavior, which in turn led to a greater number of direct violations in prison. Thus, not only do we replicate major findings in the literature that use community samples, but we also provide a mechanistic explanation for how ETV might lead to maladaptive social response patterns within their current context (**page 11**). As noted above, we comment on the robustness of the ETV results (**page 11-12**) and the predictive utility of the ETV-harm learning association for prison behavior (**page 12**),

Second, the Reviewer states “they have committed serious crimes.” We would like to echo in this response and remind the Reviewer that what really seems to differentiate these individuals is getting caught, charged and sentenced. We would like to draw the Reviewer’s attention to public reports which provide statistics on the rates of prosecution for serious crimes²⁰. The difference in the severity of criminal offenses between the incarcerated and non-incarcerated may be smaller than presumed. This FBI report states that nearly 40% of murders, 50% of aggravated assaults, 65% of rapes, and 70% of robberies are **not** cleared by arrest or other means in the year 2017. Notably, there is variance in the rates of being cleared according to region. These data highlight the reality that, in the United States, people can commit serious crimes and never be caught, charged or sentenced. Moreover, assignment to maximum security prisons is not solely based on type of crime but, given the overcrowding in the United States prison system, can be a reflection of available space and programming. Overall, these data argue against the intuition that incarcerated individuals are the only ones who have committed serious crimes.

Finally, the Reviewer intimates that incarcerated individuals will “show differences in cognition, emotion and social interaction” even when freed. Of course, some individuals who end up in prison are likely to show persistent differences in cognition, emotion, and behavior. For example, it is more likely they will engage in antisocial behavior^{21,22}. It is more likely they will continue to be exposed to violence^{13,23}. It is more likely they will continue to have experiences that are punitive and discriminatory^{24,25}. These are exactly the differences that shape how these individuals view the world. And importantly, these are exactly the experiences that make studying this population so important if we want to understand the influence of exposure to violence on behavior.

Despite all of this, there are a lot of ways incarcerated individuals are not generally different than other populations in terms of cognition, emotion and social interaction. We refer this Reviewer to their initial review, where they suggested that incarcerated individuals may differ from non-incarcerated individuals in moral attitudes, risk taking, personal relationships, social abilities, or hormone levels. In our response, we provided observational and empirical studies refuting the basic premise that those currently incarcerated individuals are fundamentally different from non-incarcerated individuals (from similar neighborhoods, of similar characteristics) ²⁻⁶. This work shows that those who end up incarcerated versus those who do not appear more similar than dissimilar on factors such as risk taking, personality traits, and mental health outcomes.

We also would like to reiterate Dr. Baskin-Sommers' findings that were presented in our initial response letter, where an identical set of self-report measures was collected from our sample of incarcerated individuals and a sample of non-incarcerated individuals from a community enriched for antisocial behavior. From this larger battery, on several measures related to the present study, **incarcerated and non-incarcerated individuals did not differ**. However, the samples did significant differ on family income (in the past year) and several components of **exposure to violence**. Notably, **the prison sample captures a full range of scores on ETV measures, reducing the issue of skew and representativeness that would be present in a community sample**. What this shows is that **in order to meaningfully test how exposure to violence affects social learning and behavior, incarcerated individuals are an ideal study population**. To make sweeping claims about fundamental differences between incarcerated versus non-incarcerated individuals without data is misguided and perpetuates an overgeneralization that vilifies the incarcerated. Being incarcerated is certainly about the person's behavior, but also about the social landscape. In this revision we clarify why examining exposure to violence in a prison sample is useful (**page 3**) and note the limitations of the sample (**page 13**).

- 2. Thus, while I agree studying prisoners is important, it is still the case that one cannot make general conclusions about moral attitudes and behaviour from studying a group of subjects living in prison. The paper, including the title ("Exposure to community violence affects the development of subjective moral impressions and trust behavior"), part of the abstract and the conclusions still make general claims that are not supported by the data. As an example I cite the first paragraph of the conclusions, in which there is no mention to the specialized sample.**

In light of the Reviewer's comments, we thoroughly reviewed each and every sentence of the revision to ensure that we do not make general claims that are not supported by the data. Additionally, we now emphasize in the title, abstract, introduction, and discussion that our findings are regarding a sample of incarcerated male inmates.

Title:

Exposure to community violence affects the development of subjective moral impressions and trust behavior in a sample of incarcerated males

Abstract:

*Exposure to community violence is a reliable predictor of negative life outcomes (e.g., problems with health, mental health, chronic aggression). Notably, **individuals exposed to violence are more likely to engage in antisocial***

behavior and, as a result, exposure to violence dramatically increases the likelihood of involvement in the justice and social service systems.

*Theoretical accounts suggest that disruptions in learning underlie the link between exposure to violence and maladaptive social behaviors (e.g., aggression, **antisocial behavior**). However, empirical evidence specifying these processes is sparse. Here, we investigated how exposure to violence affects the ability to learn about the harmfulness of others and use this information to adaptively modulate trust behavior **in a sample of currently incarcerated males. Participants** predicted the choices of two agents who repeatedly decided whether to inflict painful electric shocks on another individual in exchange for money. The agents differed substantially in their harmfulness, in that the “good” agent required more compensation to harm than the “bad” agent. Participants periodically rated their subjective impressions of the agent’s moral character, as well as their certainty of their impressions. After completing the learning task, we assessed how participants interacted with each agent in a one-shot trust game. Results indicated that exposure to violence did not impact the ability to accurately develop beliefs about the agents’ harm preferences and predict their choices. However, exposure to violence disrupted the ability to form moral impressions that dissociated between agents with distinguishable harm preferences. Consequently, **participants** with higher exposure to violence had more difficulty adjusting their trust behavior towards the two different agents. Our findings reveal a novel cognitive process that may explain the emergence of maladaptive behavior related to exposure to violence.*

We ensured that each and every paragraph of the introduction references incarcerated individuals and/or the judicial system. By framing the introduction this way, we hope to make clear to the readers (a) the direct relationship between exposure to violence and involvement in the justice and social service systems, and (b) why we embarked on the task of collecting data in a difficult to reach population such as incarcerated individuals. We made similar efforts in the discussion, and in each paragraph have referenced the association between the present findings, incarceration, and those affected by incarceration.

Additionally, throughout the manuscript, we toned down our claims to reflect our sample of incarcerated individuals. We ensure that readers understand that all results regard our *participants* and general statements about population effects directly regard *incarcerated individuals*. For example, in the first paragraph of the conclusions we explicitly emphasize that our results are within a sample of currently incarcerated males. We also removed general statements of effects related to “people” and replace this with “participants” to indicate that these findings are specifically related to the participants in our sample.

The ability to infer other’s intentions and predict their behavior is crucial for successful social interactions. In particular, learning whether others are likely to harm us is important for consequential social decisions like deciding whom to trust. However, there are environmental experiences that may impact how we learn about harm and use this information to make adaptive social decisions. Exposure to violence is one environmental experience that is associated with aberrations in beliefs about harm^{11,26}. As a result, exposure to violence is related to behaviors that reflect a lack of trust and prosociality (e.g., aggression, crime) increasing contact with systems of social control.

The current data suggest that, in a sample of currently incarcerated males, exposure to violence adversely impacts some components of harm learning, but not all. Participants with higher ETV scores showed an ability to develop accurate beliefs about others by objectively encoding their harm preferences. However, exposure to violence appeared to disrupt the formation of subjective, global impressions of other's moral character from observed harm behavior. Participants with higher ETV scores formed more positive and less uncertain impressions of harmful agents and more negative and less certain impressions of helpful agents. Moreover, these differences in subjective impressions associated with higher ETV scores led to maladaptive trust behavior, such that participants with higher ETV scores extended less trust than optimal when interacting with a "good" agent. Finally, the link between exposure to violence and maladaptive trusting behavior was mediated by the disturbances in impression formation. In turn, this led to significantly more violations in prison, suggesting that the effects of ETV on real social behavior in prison is predicted by subjective impressions and trust behavior as measured by our task. On the whole, these findings raise the intriguing possibility that exposure to violence does not fundamentally disrupt all components of social learning, but instead may produce a problem with generating global subjective social impressions and translating those impressions into adaptive social decision-making.

3. A general claim cannot be made by studying a population living in an environment that is fundamentally different. And so my initial recommendation is the same – either conduct a replication study on non-prisoners or write a paper about prisoners.

In our original response letter, we articulated several reasons for why a second study in non-prisoners is not scientifically warranted. What we did not mention is that running such a study would be resource prohibitive (aside from scientifically unwarranted). As mentioned in our response to Reviewer 1, point 2, although we successfully have run our harm learning task in online samples previously (Siegel et al., 2018), it is not possible to test the hypotheses regarding exposure to violence in the current paper in an online sample. This is because online samples do not show the full range of ETV scores observed in an incarcerated sample. Indeed, this is precisely why we ran our study in incarcerated offenders in the first place. Thus, even if we collected an extremely large online sample (which would cost several thousand dollars, given the length of our task), we would still not observe the range of ETV scores that would be necessary to compare with our current sample. The alternative -- collecting data in the lab with participants recruited from the local community -- is equally unfeasible. Previous efforts of this nature run in the senior author's lab have required two full-time research assistants, more than a year of data collection, and tens of thousands of dollars. We do not think it is appropriate or feasible to replicate our findings in a non-incarcerated sample, because such a sample would lack the range of exposure to violence necessary to make meaningful comparisons. This is why we originally embarked on the much more arduous task of collecting such data in an incarcerated sample.

Given recent efforts to encourage psychologists to run studies in populations more diverse than University students and online convenience samples, we believe our study makes an impactful contribution to the literature. It uses a sample that represents meaningful variation in the individual difference of interest (exposure to violence),

without the confound of global differences in harm learning (i.e., main effects replicate previous research). In the revised manuscript, we took great efforts to be clear that the sample was prisoners but that the research question and tasks are not about prisoners. Our research question regards the effects of exposure to violence on harmfulness learning, and because exposure to violence is well-represented in a prison sample, this makes it an ideal sample to address our research question in a meaningful way. We made substantial changes throughout the manuscript to emphasize why the sample was used, the considerations for replication in a non-incarcerated sample, and the need for future research testing generalizability.

References:

1. Siegel, J. Z., Mathys, C., Rutledge, R. B. & Crockett, M. J. Beliefs about bad people are volatile. *Nat. Hum. Behav.* **2**, 750 (2018).
2. Casciano, R. & Massey, D. S. Neighborhoods, employment, and welfare use: Assessing the influence of neighborhood socioeconomic composition. *Soc. Sci. Res.* **37**, 544–558 (2008).
3. Garbarino, J. & Sherman, D. High-Risk Neighborhoods and High-Risk Families: The Human Ecology of Child Maltreatment. *Child Dev.* **51**, 188–198 (1980).
4. Goffman, A. *On the Run: Fugitive Life in an American City*. (Picador, 2015).
5. Monahan, K. C., King, K. M., Shulman, E. P., Cauffman, E. & Chassin, L. The effects of violence exposure on the development of impulse control and future orientation across adolescence and early adulthood: Time-specific and generalized effects in a sample of juvenile offenders. *Dev. Psychopathol.* **27**, 1267–1283 (2015).
6. Tangney, J. P., Stuewig, J. & Mashek, D. J. Moral Emotions and Moral Behavior. *Annu. Rev. Psychol.* **58**, 345–372 (2007).
7. Baskin, D. & Sommers, I. Trajectories of Exposure to Community Violence and Mental Health Symptoms Among Serious Adolescent Offenders. *Crim. Justice Behav.* **42**, 587–609 (2015).
8. Fowler, P. J., Tompsett, C. J., Braciszewski, J. M., Jacques-Tiura, A. J. & Baltes, B. B. Community violence: A meta-analysis on the effect of exposure and mental health outcomes of children and adolescents. *Dev. Psychopathol.* **21**, 227–259 (2009).
9. Javdani, S., Abdul-Adil, J., Suarez, L., Nichols, S. R. & Farmer, A. D. Gender Differences in the Effects of Community Violence on Mental Health Outcomes in a Sample of Low-Income Youth Receiving Psychiatric Care. *Am. J. Community Psychol.* **53**, 235–248 (2014).
10. DuRant, R. H., Pendergrast, R. A. & Cadenhead, C. Exposure to violence and victimization and fighting behavior by urban black adolescents. *J. Adolesc. Health* **15**, 311–318 (1994).
11. Guerra, N. G., Huesmann, L. R. & Spindler, A. Community Violence Exposure, Social Cognition, and Aggression Among Urban Elementary School Children. *Child Dev.* **74**, 1561–1576 (2003).
12. Hawkins, J. D. *et al.* Predictors of Youth Violence. *Juvenile Justice Bulletin*. (2000).
13. Baskin, D. & Sommers, I. Exposure to Community Violence and Trajectories of Violent Offending. *Youth Violence Juv. Justice* **12**, 367–385 (2014).

14. Finkelhor, D., Turner, H., Ormrod, R. & Hamby, S. L. Trends in childhood violence and abuse exposure: evidence from 2 national surveys. *Arch. Pediatr. Adolesc. Med.* **164**, 238–242 (2010).
15. Fitzpatrick, K. M. & Boldizar, J. P. The prevalence and consequences of exposure to violence among African-American youth. *J. Am. Acad. Child Adolesc. Psychiatry* **32**, 424–430 (1993).
16. Hetey, R. C. & Eberhardt, J. L. The Numbers Don't Speak for Themselves: Racial Disparities and the Persistence of Inequality in the Criminal Justice System
,
The Numbers Don't Speak for Themselves: Racial Disparities and the Persistence of Inequality in the Criminal Justice System. *Curr. Dir. Psychol. Sci.* **27**, 183–187 (2018).
17. Gorman-Smith, D. & Tolan, P. The role of exposure to community violence and developmental problems among inner-city youth. *Dev. Psychopathol.* **10**, 101–116 (1998).
18. Aisenberg, E. & Herrenkohl, T. Community violence in context: risk and resilience in children and families. *J. Interpers. Violence* **23**, 296–315 (2008).
19. Every Second. *Every Second* Available at: <https://everysecond.fwd.us>. (Accessed: 2nd February 2019)
20. Clearances. *FBI* Available at: <https://ucr.fbi.gov/crime-in-the-u.s/2017/crime-in-the-u.s.-2017/topic-pages/clearances>. (Accessed: 1st February 2019)
21. Moffitt, T. E. Adolescence-Limited and Life-Course-Persistent Antisocial Behavior: A Developmental Taxonomy. *Biosocial Theories of Crime* (2017). doi:10.4324/9781315096278-3
22. Mulvey, E. P. *et al.* Theory and Research on Desistance from Antisocial Activity among Serious Adolescent Offenders. *Youth Violence Juv. Justice* **2**, 213–236 (2004).
23. Fagan, A. A. The Relationship Between Adolescent Physical Abuse and Criminal Offending: Support for an Enduring and Generalized Cycle of Violence. *J. Fam. Violence* **20**, 279–290 (2005).
24. Wilson, W. J. *More Than Just Race: Being Black and Poor in the Inner City (Issues of Our Time)*. (W. W. Norton & Company, 2009).
25. Goff, P. A., Jackson, M. C., Di Leone, B. A. L., Culotta, C. M. & DiTomasso, N. A. The essence of innocence: consequences of dehumanizing Black children. *J. Pers. Soc. Psychol.* **106**, 526–545 (2014).
26. Ng-Mak, D. S., Stueve, A., Salzinger, S. & Feldman, R. Normalization of Violence Among Inner-City Youth: A Formulation for Research. *Am. J. Orthopsychiatry* **72**, 92–101 (2002).

****REVIEWERS' COMMENTS:**

Reviewer #1 (Remarks to the Author):

The revised manuscript clearly addresses the issues I raised in previous rounds of review and I am now pleased to recommend this article for publication.